# HadSkip: Homotopic and Adaptive Layer Skipping of Pre-trained Language Models for Efficient Inference

**Haoyu Wang**[§]**, Yaqing Wang**[†]**, Tianci Liu**[§]**, Tuo Zhao**[*]**, and Jing Gao**[§]

[§]Purdue University, West Lafayette, IN, USA
[†]Google Research, New York, NY, USA
[*]Georgia Institute of Technology, Atlanta, GA, USA
[§]{wang5346,liu3351,jinggao}@purdue.edu,
[†]yaqingwang@google.com,
[*]tourzhao@gatech.edu

## Abstract

Pre-trained language models (LMs) have brought remarkable performance to numerous NLP tasks. However, they require significant resources and entail high computational costs for inference, making it challenging to deploy them in real-world and real-time systems. Existing early exiting methods aim to reduce computational complexity by selecting the layer at which to exit, but suffer from the limitation that they have to sequentially traverse through all layers prior to the selected exit layer, which lacks flexibility and degrades their performance. To solve this problem, we propose a **h**omotopic and **ad**aptive layer **skip**ping fine-tuning method named HadSkip. HadSkip adaptively selects the layers to skip based on a predefined budget. Specifically, we introduce a learnable gate before each layer of the LM to determine whether the current layer should be skipped. To tackle various challenges in training brought by discrete gates and budget constraints, we propose a fine-grained initialization strategy and homotopic optimization strategy. We conduct extensive experiments on the GLUE benchmark, and experimental results demonstrate the proposed HadSkip outperforms all state-of-the-art baselines significantly.

## 1 Introduction

Pre-trained language models (LMs), such as BERT (Devlin et al., 2018), RoBERTa (Liu et al., 2019), and DeBERTa (He et al., 2020), have significantly improved performance across various natural language processing (NLP) tasks, including paraphrase identification (Wang et al., 2021), natural language inference (Jiang and de Marneffe, 2019), sentiment classification (Gao et al., 2019), and so on. Pre-trained LMs are typically built by stacking transformer (Vaswani et al., 2017) layers or their variants, where the self-attention mechanism within each layer often exhibits high computational complexity. Consequently, the inference complexity of pre-trained LMs has been a bottle-neck for deploying them in latency-sensitive or latency-constrained scenarios (Xin et al., 2020).

To comply with the demand for inference-efficient models, several approaches have been proposed to accelerate pre-trained LM inference. These methods include weight quantization (Zafrir et al., 2019; Kim et al., 2021), pruning (Liu et al., 2021), knowledge distillation (Sanh et al., 2019; Jiao et al., 2019), and early exiting (Xin et al., 2020; Zhou et al., 2020; Xin et al., 2021). Among them, early exiting methods become more appealing as they do not rely on specific hardware support, such as custom chips (Liu et al., 2023) or GPUs, and they do not require training an efficient model from scratch on large pre-training corpora, thus providing a more cost-effective solution. Early exiting methods attach a classifier to each transformer layer and aim to determine if the model should exit (stop inference) from the current layer based on a criterion or learned decision. Both the backbone model and the classifiers are jointly trained in the fine-tuning stage.

Although early exiting methods enable the selection of a layer to exit based on the input sequence, they still require the input to pass through the transformer layers sequentially before exiting. This prevents the model from choosing desired uncontiguous layers. For example, with a 2-layer budget, for some input sequences, choosing the second and fifth layers of the model could achieve the most superior performance. However, early exiting methods can only traverse the layers sequentially and exit at most at the second layer, thereby missing out on the optimal combination of the second and fifth layers and potentially leading to degraded model accuracy as a result of suboptimal layer selection.

To address this issue, we propose a **h**omotopic and **ad**aptive layer **skip**ping fine-tuning method named HadSkip. The proposed HadSkip selects which layers can be skipped based on the difficulty of input sequences and pre-defined budget.

Specifically, we introduce a binary learnable gate before each transformer layer of the LM to control whether the model can skip the current layer. This allows the model to use different gate values for different input sequences, satisfying the inference budget requirements. However, learning binary gates is a non-trivial task. The binary gates are discrete and it is difficult to optimize by gradient-based optimizers directly. Besides, the presence of a budget constraint can lead to early optimization collapse, particularly when the budget value is small. To tackle them, we design an elaborated three-stage training strategy to tackle this challenge. The first stage finetunes the backbone model and uses it to initialize the transformer layers. Next, these transformer layers are frozen and the gate parameters are initialized. In the third stage, the transformer layers and gates are trained jointly. To compute the gradient of gates, we employ a second-order gradient approximation method ReinMax (Liu et al., 2023). Further, we propose a homotopic optimization strategy, which converts the optimization process into a sequence of problems ranging from a large budget to a small budget. On one hand, this ensures a smoother and more stable optimization process. On the other hand, it avoids skipping a large number of layers at the beginning of training and losing valuable information from the pre-trained model.

The contributions of the paper are summarized as follows: 1) We propose a homotopic and adaptive layer skipping fine-tuning method to accelerate the inference of pre-trained LMs. It is more flexible and economical compared to existing LM inference acceleration methods. 2) We design a three-stage and homotopic optimization strategy. This strategy makes the optimization more stable and easier to solve. 3) We conduct extensive experiments on the GLUE benchmark. Results show that the proposed HadSkip outperforms baselines significantly and achieves a good efficiency-accuracy trade-off, e.g., preserving 95.7% of BERT's performance with only half the layers on average.

## 2 Related Work

Existing approaches for accelerating the inference of pre-trained language models can be broadly categorized into two types: 1) model compression-based methods and 2) early exiting based methods. The proposed method is more relevant to early exiting based methods, which are discussed in more detail as follows. The discussion about the model compression-based methods can be found in the appendix.

Early exiting methods enable the production of inference-efficient models for specific inference budgets during the fine-tuning stage. These methods draw inspiration from adaptive computation (Graves, 2016) techniques in recurrent neural networks and BranchyNet (Teerapittayanon et al., 2016) in computer vision. The fundamental concept behind early exiting methods is to identify the layer at which to exit early, rather than sequentially passing through all layers. Based on the criteria of choosing the exiting layer, they can be organized into three strategies. The first is score-based early exiting. (Liu et al., 2020; Xin et al., 2020; Kaya et al., 2019) use the entropy of the prediction probability and the maximum of the predicted distribution as the score for exit determination respectively. The second is learning to exit. BERxiT (Xin et al., 2021) used a fully connected layer right after each layer to produce decisions if the model needs to exit from the current layer. The third is patience-based early exiting. For example, PABEE (Zhou et al., 2020), SENTEE (Li et al., 2021) and LeeBERT (Zhu, 2021) were designed to use scores of multiple layers to determine if the model could exit. Recently, PCEE-BERT (Zhang et al., 2022) also combined the score-based method with the patience-based early exiting method to improve accuracy. Despite these advancements, early exiting methods still require sequential traversal through multiple layers. However, for many input sequences, the optimal layers for inference could be non-contiguous. This lack of flexibility in early exiting methods can limit their performance. To address this limitation, inspired by BlockDrop (Wu et al., 2018), we propose a homotopic and adaptive layer skipping fine-tuning method named HadSkip, which can dynamically select which layers can be skipped based on the difficulty of input sequences and the pre-defined budget. Concurrently, Smart-BERT (Hu et al., 2023) proposes to combine layer skipping and early exiting based on cross-layer contrastive learning.

## 3 Preliminaries

In this section, we first introduce the architecture of the pre-trained language model (PLM) and analyze its complexity for inference.

### 3.1 Pre-trained Language Model

A pre-trained language model usually consists of embedding layers, multiple transformer layers, and an output layer. Specifically, for an input sequence

$x$, we obtain its corresponding embedding first:

$$\boldsymbol{e} = \boldsymbol{e}_{token}(x) + \boldsymbol{e}_{pos}(x) + \boldsymbol{e}_{seg}(x),$$

where $\boldsymbol{e}_{token}$, $\boldsymbol{e}_{pos}$, and $\boldsymbol{e}_{seg}$ are the token embedding layer, position embedding layer and segment embedding layer respectively. Then the embedding $\boldsymbol{e}$ will be fed into transformer layers, which can be formulated as $\boldsymbol{h}_0 = \boldsymbol{e}$ and

$$\boldsymbol{h}_i = f_i(\boldsymbol{h}_{i-1}; \theta_i), i = 1, 2, ..., L, \qquad (1)$$

where $h_i$, $\theta_i$, and $f_i$ are the output, parameters, and mapping function of the $i$-th transformer layer respectively. In the end, the output of the last transformer layer passes the output layer and generates logits, which can be represented as $\boldsymbol{o} = f_{out}(\boldsymbol{h}_L)$.

### 3.2 Inference Complexity

The inference complexity consists of the complexity of three parts: 1) embedding layers, 2) transformer layers, and 3) the output layer. The time complexity of both the embedding layers and the output layer is approximately $\mathcal{O}(d^3)$. The complexity of transformer layers is $\mathcal{O}(4md^2L + m^2dL)$, where $m$ represents the sequence length, $d$ denotes the embedding dimensionality, and $L$ signifies the number of transformer layers. Based on the analysis, it is evident that the complexity of transformer layers significantly exceeds that of the embedding layers and the output layer. Hence, this paper proposes to skip some transformer layers to reduce inference complexity, leveraging the expected number of transformer layers as a budget.

## 4 Methodology

Given an $L$-layer pre-trained language model (PLM) and a training set $\mathcal{D} = \{(x^i, y^i)\}_{i=1}^n$, where $x^i$ is the input sequence, $y^i$ is the corresponding label and $n$ is the number of training data, the aim is to train a model $f(x^i) \to y^i$ which can be efficient for inference while preserving the model performance.

### 4.1 Overview

To enhance model inference efficiency, we propose a **h**omotopic and **ad**aptive layer **skip**ping finetuning method named HadSkip, which is shown in Fig. 1. The motivation behind HadSkip is to reduce the complexity of transformer layers by adaptively skipping certain layers based on the input sequence. We introduce gated transformer layers in which a binary gate is incorporated into each transformer layer to determine whether the current layer should be skipped (Section 4.2). Additionally, to address the non-differentiable nature of the binary gates and

meet the budget contraint, we propose a three-stage optimization strategy in Section 4.3.

### 4.2 Gated Transformer Layer

Based on the analysis in Section 3.2, we identify that the transformer layers are the bottleneck in terms of inference complexity. Since the difficulty of input sequences can vary due to factors such as vocabulary, sequence length, and rhetoric, it is unnecessary to utilize all $L$ transformer layers for every input sequence. Using a smaller number of transformer layers may already yield accurate prediction results for certain simple sequences. Therefore, motivated by (Wu et al., 2018), we propose a gated transformer layer that adaptively selects layers to enhance inference efficiency.

Specifically, we introduce a binary gate before feeding input into each transformer layer. This gate determines whether the current transformer layer should be bypassed based on the output of the previous layer. Formally, the forward propagation can be represented as

$$\boldsymbol{h}_0 = \boldsymbol{e}, \boldsymbol{h}_i = f_i(\boldsymbol{h}_{i-1}; \theta_i) \cdot g_i + \boldsymbol{h}_{i-1} \cdot (1 - g_i),$$

where $g_i = \mathcal{B}(g(\boldsymbol{h}_{i-1}; \omega_i)) \in \{0, 1\}$ is the binary gate, $\mathcal{B}(\cdot) : \mathbb{R} \to \{0, 1\}$ is a binarized function, $g(\cdot)$ is a one hidden layer feed-forward network, and $\omega_i$ is the gate parameter. If the value of gate $g_i$ is 1, then $\boldsymbol{h}_i = f_i(\boldsymbol{h}_{i-1}; \theta_i)$, indicating that the current $i$-th transformer layer is preserved; otherwise, it is skipped.

### 4.3 Optimization Strategy

Considering a language model composed of stacked gated transformer layers, we need to impose a constraint on the maximum number of used layers, which should be approximately equal to a specified budget. The optimization problem can be formally defined as follows:

$$\min_{\Theta} \frac{1}{n} \sum_j \ell(f(x^j; \Theta))$$

$$\text{s.t.} \mathbb{E}[g_i] \approx \frac{1}{nL} \sum_{j=1}^{n} \sum_{i=1}^{L} g_i^j = s, \qquad (2)$$

where $\Theta$ is the collection of model parameters consisting of transformer layer parameters, gate parameters, embedding layer parameters and output layer parameters, $g_i^j$ is the gate of $i$-th transformer layer with respect to input sequence $x^j$, $\ell(\cdot)$ is the loss function, the specific form of which will be demonstrated in Section 4.3 in detail, and $s$ is the pre-given budget. The mean square error is employed as a penalty term in the problem in Eqn. 2,

resulting in the following formulation:

$$\min_{\Theta} \frac{1}{n} \sum_j \ell(f(x^j; \Theta)) + \beta \left( \frac{1}{nL} \sum_{j=1}^{n} \sum_{i=1}^{L} g_i^j - s \right)^2, \quad (3)$$

where $\beta$ is a hyper-parameter. However, because of the non-differentiability of $g_i^j$, the optimization process can easily converge to a suboptimal solution, making it challenging to solve directly. Additionally, due to the budget constraint, the model might prioritize skipping a significant number of layers first to meet the constraint. This behavior may lead to unstable optimization, and a loss of knowledge learned during the pretraining phase, which is usually stored in the first several layers of the model. To handle this problem, we propose a three-stage optimization strategy. In the initial two stages, we aim to find a good initialization for the HadSkip, while in the third stage, we propose the utilization of a homotopic optimization method, which enables learning of the gate values. The subsequent sections will provide an introduction to each of these three stages.

● **Stage I: Initialize Transformer Layer Parameters**. Since transformer layer parameters are real-valued, they are generally easier to optimize compared to gate parameters. The discrepancy in optimization difficulty poses a challenge for model convergence. Motivated by this observation, we propose separate initialization design for transformer layer parameters and for gate parameters respectively. This section focuses on the initialization of transformer layer parameters. Specifically, we restrict our consideration to a vanilla language model $f_v(\cdot)$ comprising transformer layers in order to avoid non-differentiable operations. The training process for $f_v$ involves standard fine-tuning of a language model, which can be formulated as

$$\min_{\Theta_v} \frac{1}{n} \sum_j \ell_t(f_v(x^j; \Theta_v), y^j), \quad (4)$$

where $\ell_t(\cdot, \cdot)$ denotes the task-specific loss function, such as cross-entropy for classification, and $\Theta_v$ represents the parameters of $f_v$, comprising transformer layer parameters, embedding layer parameters, and output layer parameters. The learned $\Theta_v$ is utilized to initialize the corresponding parameters of the model $f$, denoted as $\Theta \backslash \{\omega_i\}_{i=1}^{L} \leftarrow \Theta_v$.

● **Stage II: Initialize Gate Parameters**. Following the initialization of transformer layer parameters, we freeze the parameters $\Theta \backslash \{\omega_i\}_{i=1}^{L}$ and relax the original optimization problem in Eqn. 2 to achieve a coarse-grained initialization for gate parameters.

The expectation constraint is omitted and we solve the following optimization problem:

$$\min_{\{\omega_i\}_{i=1}^{L}} \frac{1}{n} \sum_j \ell(f(x^j; \Theta)).$$

To handle the non-differentiable problem, previous work usually leverages straight-through estimation (STE) (Bengio et al., 2013) to obtain approximate gradients. However, we observe that STE, although a first-order approximation (Liu et al., 2023), still presents challenges in stable backpropagation and may result in model collapse. To address this issue, we employ a second-order gradient approximation method called ReinMax (Liu et al., 2023). It integrates Heun's Method to approximate the gradient of $\mathcal{B}(\cdot)$, which is shown in Algorithm 1 in the Appendix.

● **Stage III: Train LM and Gate Jointly**. Following the completion of the first two stages, the proposed model has achieved a favorable initialization, and the gradient estimation is stable. However, in cases where the budget is limited and requires the LM to skip multiple layers, the substantial difference in model capacity leads to significant performance degradation. To alleviate this problem, we propose a homotopic optimization strategy that incorporates knowledge distillation. The knowledge distillation borrows knowledge from a fine-tuned vanilla LM to guide the proposed HadSkip learning via regularizing the consistency between their hidden representation and output predictions. Additionally, the homotopic optimization strategy utilizes a greedy method to facilitate smoother optimization.

● *Knowledge Distillation*. We incorporate the following distillation losses into consideration: 1) the distillation loss of prediction $\mathcal{L}_p$, 2) the distillation loss of hidden representations $\mathcal{L}_h$, 3) the distillation loss of attention $\mathcal{L}_{att}$, and 4) the distillation loss of embedding layer $\mathcal{L}_{emb}$. Further elaboration on the distillation losses and the total loss can be found in Section B in the appendix.

● *Homotopic Optimization Strategy*. While leveraging initialization and knowledge distillation can enhance the performance of HadSkip, optimizing the model remains challenging due to the significant capacity and representation power disparity between $f$ and $f_v$. To address this issue, we propose a homotopic optimization strategy that transforms the optimization into a sequence of problems, gradually progressing from easier to more challenging ones. Formally, we solve a series of optimiza-

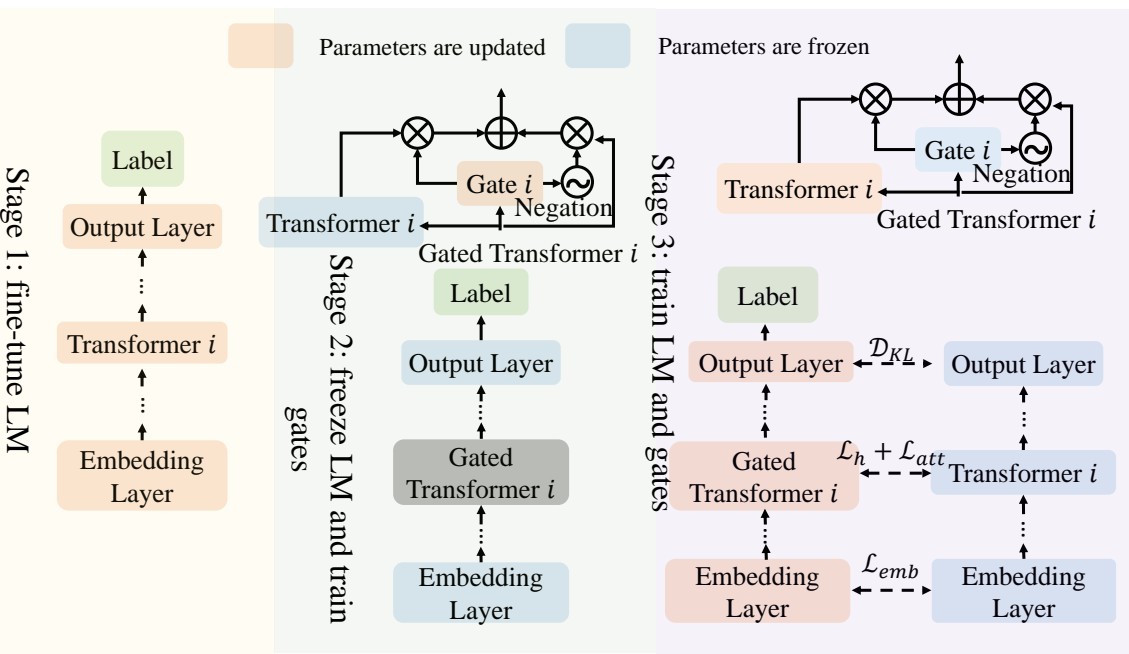

Figure 1: The framework of HadSkip.

tion problems by replacing $s$ with $k$ in Eqn. 3 for $k = s_0, s_1, \ldots, s_q$, where $s_0 > s_1 > \ldots > s_q = s$. Hence, the result of the last optimization becomes the initialization for the subsequent optimization problem. Here, we observe that Eqn. 3 is not highly sensitive to the choice of $\beta$. Therefore, we employ a uniform $\beta$ value across all optimization problems. Additionally, we find seeting $s_0 = L, s_1 = L - 1, \ldots, s_q = s$ is a simple yet effective choice.

## 5  Experiment

In this section, we evaluate the proposed HadSkip and answer the following questions: RQ1) How does HadSkip perform compared to state-of-the-art early exiting baselines? RQ2) What are the roles of initialization, knowledge distillation, and homotopic optimization strategy in model performance improvements respectively? RQ3) Can the proposed HadSkip generalize well with respect to different backbones? RQ4) Is it possible to combine the proposed HadSkip method with other model acceleration techniques? RQ5) How does the performance change with varying budget values? RQ6) Can the proposed HadSkip capture the difficulty differences among input sequences?

### 5.1  Datasets and Experiment Settings

• **Datasets** Experiments are conducted on the benchmark GLUE dataset following the methodology of (Zhang et al., 2022; Sun et al., 2019). Specifically, we evaluate our approach on seven classification tasks from the GLUE benchmark, namely MRPC, SST-2, RTE, QQP, QNLI, MNLI, and CoLA, which use F1, Accuracy, Accuracy, F1, Accuracy, Accuracy, and Matthews correlation coefficient as metrics respectively (the higher the better). More details about the datasets and evaluation metrics are provided in Table 5 in the Appendix. The performance is reported on the development sets, following the approach of (Zhang et al., 2022; Sun et al., 2019).

• **Baselines** We adopt two types of baselines: 1) vanilla pre-trained language model BERT (Devlin et al., 2018), and 2) early exiting methods, including budgeted exiting (Zhang et al., 2022), BranchyNet (Teerapittayanon et al., 2016), Shallow-Deep (Kaya et al., 2019), BERxiT (Xin et al., 2021), PABEE (Sun et al., 2019), and PCEE-BERT (Zhang et al., 2022). Since BERT does not have an inference speedup design, it serves as the performance ceiling for inference speedup methods. Budgeted exiting uses top $K$ layers to make predictions based on the budget. BranchyNet, Shallow-Deep, BERxiT, PABEE, and PCEE-BERT dynamically select a layer to exit according to the input sequence. Following (Zhang et al., 2022), we make early exiting baselines and proposed HadSkip expectedly use 3, 6, and 9 layers as budgets in experiments. Unless otherwise specified, all these methods utilize BERT as the backbone.

| | CoLA | MNLI | MRPC | QNLI | QQP | RTE | SST-2 | AVG |
|---|---|---|---|---|---|---|---|---|
| BERT* | 59.4 | 84.3 | 91.3 | 88.5 | 90.4 | 69.3 | 92.5 | 82.2 |
| Budgeted-Exit* | 0.0 | 70.0 | 75.8 | 77.4 | 81.8 | 54.7 | 81.0 | 63.0 |
| BranchyNet* | 0.0 | 63.8 | 75.7 | 74.2 | 71.6 | 54.7 | 79.9 | 60.0 |
| Shallow-Deep* | 0.0 | 64.1 | 75.6 | 74.3 | 71.4 | 54.7 | 79.5 | 59.9 |
| BERxiT* | 0.0 | 63.5 | 75.6 | 73.3 | 68.2 | 55.3 | 79.5 | 59.3 |
| PABEE* | 0.0 | 63.9 | 75.8 | 73.6 | 68.6 | 55.8 | 79.9 | 59.7 |
| PCEE-BERT* | 9.8 | 73.4 | 78.8 | 80.4 | 79.6 | 58.4 | 83.6 | 66.3 |
| HadSkip-BERT | 35.6 | 78.1 | 84.2 | 84.1 | 88.5 | 54.9 | 88.5 | 73.4 |
| Budgeted-Exit* | 0.0 | 79.6 | 84.7 | 85.3 | 89.3 | 68.1 | 88.6 | 70.8 |
| BranchyNet* | 0.0 | 78.3 | 83.0 | 87.1 | 89.3 | 67.4 | 88.3 | 70.5 |
| Shallow-Deep* | 0.0 | 78.2 | 82.8 | 87.2 | 89.6 | 67.2 | 88.4 | 70.5 |
| BERxiT* | 12.3 | 78.4 | 82.9 | 87.0 | 89.1 | 67.3 | 88.3 | 72.2 |
| PABEE* | 0.0 | 78.9 | 83.1 | 87.2 | 89.6 | 67.7 | 88.7 | 70.7 |
| PCEE-BERT* | 23.2 | 80.1 | 84.8 | 87.1 | 90.8 | 69.4 | 90.4 | 75.1 |
| HadSkip-BERT | 50.8 | 82.4 | 85.9 | 88.7 | 90.5 | 62.5 | 90.0 | 78.7 |
| Budgeted-Exit* | 51.9 | 83.0 | 87.0 | 88.4 | 90.3 | 69.0 | 91.2 | 80.1 |
| BranchyNet* | 52.1 | 83.0 | 85.8 | 89.3 | 90.1 | 68.0 | 91.2 | 79.9 |
| Shallow-Deep* | 52.3 | 82.9 | 85.7 | 89.3 | 90.1 | 67.8 | 91.2 | 79.9 |
| BERxiT* | 52.2 | 83.2 | 86.2 | 89.6 | 90.1 | 68.1 | 91.4 | 80.1 |
| PABEE* | 52.4 | 83.4 | 86.1 | 89.8 | 90.4 | 68.3 | 91.7 | 80.3 |
| PCEE-BERT* | 52.8 | 83.4 | 86.8 | 90.5 | 91.2 | 69.7 | 91.8 | 80.9 |
| HadSkip-BERT | 58.3 | 83.8 | 88.4 | 90.7 | 90.8 | 66.8 | 92.7 | 81.6 |

Table 1: Performance comparison on GLUE dataset. Blue, green, and orange shadow parts represent methods with 3, 6, and 9-layer budgets respectively. "AVG" notes the average performance of the 7 tasks. Results with "*" are taken from (Zhang et al., 2022).

## 5.2 Performance Comparison

This section presents the performance of the baselines and the proposed HadSkip model, as shown in Table 1, addressing RQ1.

The performance results in Table 1 demonstrate that the proposed HadSkip surpasses all state-of-the-art baselines on the GLUE datasets. Notably, under budget configurations of 3, 6, and 9 layers, the proposed HadSkip achieves substantial improvements in terms of average accuracy. Specifically, compared to the baselines, HadSkip exhibits performance gains of at least 10.7%, 4.8%, and 0.9% when using 3, 6, and 9 layers, respectively. Additionally, when compared to vanilla BERT, the proposed HadSkip retains 89.3%, 95.7%, and 99.3% of its performance when using 3, 6, and 9 layers, respectively. This demonstrates that HadSkip effectively accelerates model inference while preserving the performance of the pre-trained language model.

We also observe that as the budget decreases, the diversity in model performance becomes more pronounced. Specifically, when using 9 layers, the performance of the baselines, apart from the proposed HadSkip, exhibits minimal variation. However, with 3 and 6 layers, the performance gaps among different methods become more prominent. The proposed HadSkip demonstrates much more significant improvements under these two settings.

Specifically, on the CoLA dataset, the baselines struggle to provide accurate classification results when using 3 and 6 layers, while the proposed HadSkip remains effective. This indicates that the proposed HadSkip exhibits greater superiority over the baselines when operating under a smaller budget. Hence, in scenarios with limited resources, selecting layers adaptively may be more effective than choosing the exit layer.

| | Budget | MRPC | MNLI | CoLA |
|---|---|---|---|---|
| HadSkip w/o gate initialization | 3 | 82.1 | 77.0 | 34.4 |
| | 6 | 86.9 | 81.5 | 50.6 |
| | 9 | 88.7 | 83.7 | 56.3 |
| HadSkip w/o homotopic optimization | 3 | 81.9 | 75.1 | 5.1 |
| | 6 | 83.1 | 81.7 | 39.9 |
| | 9 | 88.8 | 83.7 | 51.6 |
| HadSkip w/o KD | 3 | 83.0 | 62.9 | 17.9 |
| | 6 | 87.3 | 80.8 | 47.8 |
| | 9 | 88.7 | 82.7 | 57.6 |
| HadSkip only w/ prediction KD | 3 | 85.5 | 77.7 | 29.6 |
| | 6 | 86.7 | 81.7 | 48.4 |
| | 9 | 87.8 | 83.5 | 58.6 |
| HadSkip | 3 | 84.2 | 78.1 | 35.6 |
| | 6 | 85.9 | 82.4 | 50.8 |
| | 9 | 88.4 | 83.8 | 58.3 |

Table 2: Performance of HadSkip with and without the proposed gate initialization, HadSkip with and without the proposed homotopic optimization strategy, and HadSkip with and without knowledge distillation. HadSkip w/o KD represents HadSkip only uses task-specific loss. HadSkip w/ prediction KD represents HadSkip that uses both task-specific loss and the distillation loss of prediction.

## 5.3 Ablation Study

● **Effectiveness of Gate Initialization**. We conduct an analysis to answer RQ2 regarding gate initialization. To validate the role of the gate initialization, we compare HadSkip without the second stage to the complete HadSkip architecture. The comparison results are presented in Table 2. The experimental findings confirm the positive impact of gate initialization on model training. We observe that gate initialization provides greater performance improvement when the budget is small. However, for larger budgets, the benefits of gate initialization are limited. This can be attributed to the fact that with a larger budget, the model only needs to skip a few layers, which is relatively easier compared to skipping multiple layers. In such

cases, a fine-grained warm-up strategy becomes unnecessary.

| | Budget | MRPC | MNLI | CoLA |
|---|---|---|---|---|
| RoBERTa* | - | 91.7 | 87.2 | 64.8 |
| DeeBERT+RoBERTa♡ | 3 | 81.2 | 49.5 | 0.0 |
| | 6 | 81.2 | 65.0 | 0.0 |
| | 9 | 92.4 | 86.2 | 55.7 |
| HadSkip+RoBERTa | 3 | 85.1 | 81.0 | 35.3 |
| | 6 | 90.0 | 85.6 | 55.8 |
| | 9 | 91.9 | 87.2 | 60.2 |

Table 3: Performance of using RoBERTa as the backbone. Results with ⋆ are taken from (Lee et al., 2019). Results with ♡ are implemented based on its official code [1].

| | Budget | MRPC | MNLI | CoLA |
|---|---|---|---|---|
| TintBERT(4 layers)* | - | 89.1 | 80.4 | 18.6 |
| TinyBERT(6 layers) * | - | 91.6 | 83.5 | 42.8 |
| HadSkip+TinyBERT(6 layers) | 3 | 89.8 | 80.6 | 35.5 |
| | 4 | 91.3 | 82.2 | 40.5 |

Table 4: Performance of using TinyBERT as the backbone. Results with "*" are taken from (Liang et al., 2023).

● **Effectiveness of Homotopic Optimization Strategy.** We conduct an ablation study to answer RQ2 with respect to the homotopic optimization strategy. We compare the proposed HadSkip with HadSkip without the homotopic optimization strategy on the MRPC, MNLI, and CoLA datasets, as presented in Table 2. From the results in Table 2, we observe the effectiveness of the proposed homotopic optimization strategy in improving model performance. The homotopic optimization strategy leads to average improvements of 201.6%, 10.5%, and 4.2% when using 3, 6, and 9 layers, respectively. This trend highlights the greater importance of the homotopic optimization strategy for smaller budgets. When using 9 layers, the performance of HadSkip with and without the homotopic optimization strategy is similar for the MRPC and MNLI datasets. However, when using 3 layers, HadSkip without the homotopic optimization strategy fails to produce meaningful output for CoLA, whereas HadSkip still delivers competitive results. One potential reason for this difference is that the homotopic optimization strategy smooths the optimization problem and facilitates convergence to a better local optimum.

● **Effectiveness of Knowledge Distillation.** In this paragraph, we do the ablation study to answer

RQ2 regarding knowledge distillation. We conduct two comparison experiments: 1) HadSkip without knowledge distillation (KD) versus HadSkip, and 2) HadSkip using only prediction-based knowledge distillation versus HadSkip. The experimental results are presented in Table 2. Prediction-based knowledge distillation is widely employed in traditional teacher-student architectures, and the knowledge distillation loss function used in HadSkip is popular for compressing large language models. Compared to HadSkip without KD, HadSkip consistently improves model performance. For instance, HadSkip achieves improvements of 98.9%, 6.3%, and 1.2% when using 3, 6, and 9 layers, respectively. Compared to HadSkip with only prediction KD, HadSkip still shows improvements. This demonstrates that the losses of hidden representations, attention matrices, and embeddings effectively guide the model in learning how to select layers to skip. However, from Table 2, we observe that the performance improvements resulting from the distillation loss of prediction are greater than those from other distillation losses. This highlights the more significant role of the distillation loss of prediction in the proposed HadSkip.

## 5.4 Extension to RoBERTa

In this section, we employ RoBERTa (Liu et al., 2019) as the backbone to investigate RQ3. The results are presented in Table 3. We also consider DeeBERT, which utilizes RoBERTa as the backbone in their paper, as our baseline. As shown in Table 3, HadSkip+RoBERTa outperforms DeeBERT+RoBERTa. Specifically, for CoLA and MNLI, DeeBERT+RoBERTa fails to make correct predictions when using 3 and 6 layers. In contrast, the proposed HadSkip+RoBERTa achieves comparable performance to RoBERTa even with only 6 layers. These results demonstrate that HadSkip can effectively balance efficiency and accuracy when combined with different backbone models.

## 5.5 Combining with TinyBERT

In this section, we investigate RQ4 using Tiny-BERT (Jiao et al., 2019) as the backbone. Apart from early exiting methods, another popular approach to accelerate model inference is reducing the number of LM layers via knowledge distillation for LM compression. To demonstrate the complementary of the proposed HadSkip with knowledge distillation model compression methods, we utilize a 6-layer TinyBERT, which compresses BERT using knowledge distillation into a

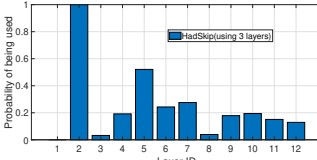 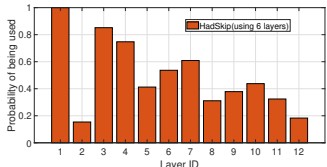 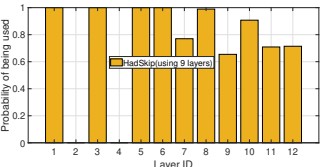

Figure 2: Visualization of the probability of each layer being used on MNLI dataset.

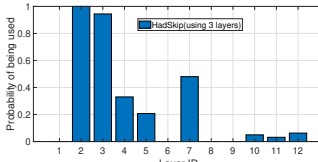 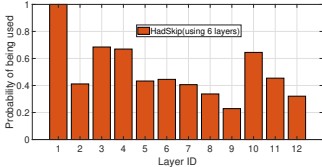 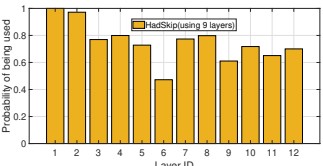

Figure 3: Visualization of the probability of each layer being used on QNLI dataset.

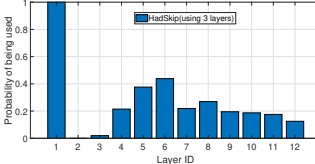 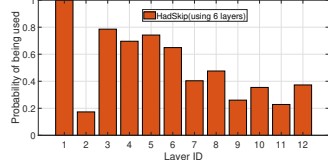 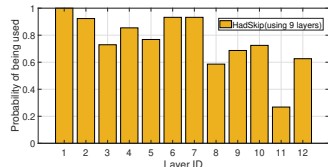

Figure 4: Visualization of the probability of each layer being used on SST2 dataset.

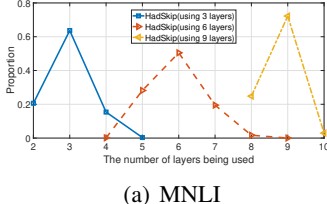 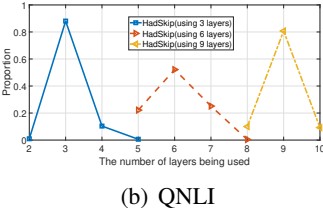 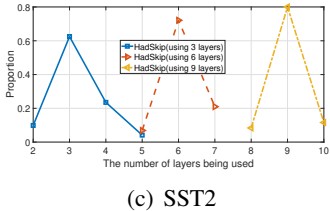

(a) MNLI       (b) QNLI       (c) SST2

Figure 5: Visualization of the probability of the number of layers being used with respect to different budgets.

6-layer small model, as the backbone and present the performance results in Table 4. Comparing HadSkip with a budget of 4 layers to TinyBERT (6 layers), we observe that HadSkip can preserve over 94.6% of the model performance. This indicates that HadSkip can effectively complement TinyBERT and further enhance the inference speed of TinyBERT. Moreover, when compared to the pre-trained 4-layer TinyBERT, HadSkip with 4 layers achieves higher accuracy, and even HadSkip with 3 layers demonstrates similar performance. These results suggest that combining HadSkip with a compressed model may yield greater effectiveness than training a smaller compression model.

## 5.6 Sensitivity w.r.t. Budgets

We perform a hyperparameter experiment to address RQ5, focusing on different budgets for HadSkip using BERT-Large, BERT-base, and RoBERTa as backbones. BERT-Large is a 24-layer language model, while BERT-base and RoBERTa consist of 12 layers each. The results are presented in Fig. 6. It is evident from the figure that larger budgets lead to better performance. However, when the budget is less than half the total number of layers, the decrease in accuracy is significantly more pronounced compared to cases where the budget exceeds half the total number of layers. Specifically, as shown in Fig. 6, using half the total number of layers still preserves over 97% of the model's performance. This finding suggests that it is preferable to skip fewer than half the total number of layers.

## 5.7 Case Study

In this section, we conduct a case study to answer RQ6. To study if the proposed HadSkip can adaptively select layers based on input sequences, we visualize 1) the probability of each layer being used and 2) the probability of the number of layers being used on MNI, QNLI and SST2 datasets. We

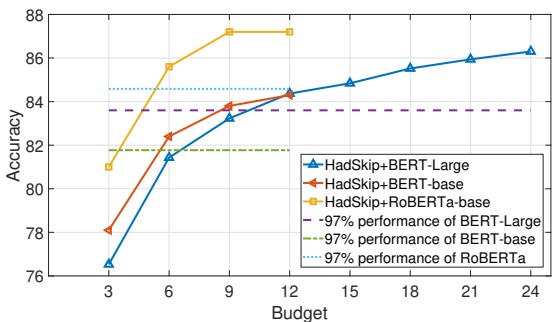

Figure 6: Accuracy with different budgets.

show the visualization on Fig. 2 to Fig. 5. From Fig. 2 to Fig. 4, we observe diverse probabilities of each layer being utilized across different tasks and budgets. When utilizing 3 layers, the model tends to prefer specific layers. For example, the model favors the second and fifth layers on the MNLI dataset and the second, third, and seventh layers on the QNLI dataset. However, as the budget increases, the distribution of chosen layers becomes more uniform. According to Fig. 5, the number of layers utilized concentrates around the given budget, while the model also employs other numbers of layers near the budget. These figures confirm that the proposed model can select different layers for various input sequences.

## 6   Conclusion

In this paper, we propose a homotopic and adaptive layer skipping fine-tuning method named HadSkip for efficient inference. We introduce a learnable gate before each transformer layer of the pre-trained language model to determine whether the model can skip the respective layer. For training the binary gate, we introduce a three-stage learning strategy to initialize and update model parameters. Additionally, we propose a homotopic optimization strategy to stabilize the model training. We also utilize knowledge distillation to guide model training and achieve an improved trade-off between efficiency and accuracy. We conducted extensive experiments on the GLUE benchmark, and the results demonstrate that HadSkip outperforms state-of-the-art baselines. Moreover, HadSkip complements other acceleration methods, including TinyBERT.

## Limitations

The proposed HadSkip introduces multiple hyper-parameters including $\beta$, which might require additional effort to tune. Fortunately, we observe that the prediction performance is not sensitive to these hyperparameters, and therefore they can be easily tuned. In this paper, we set them to fixed values (e.g., one) and achieve good results.

## Ethics Statement

This paper proposes a homotopic and adaptive layer skipping fine-tuning method, called HadSkip. We show that the proposed HadSkip can be used for inference acceleration. We perform experiments on classification tasks using the GLUE benchmark. In all experiments, we utilize public benchmark datasets, models, and code. No ethical concerns were identified.

## Acknowledgement

This work is supported in part by the US National Science Foundation under grant NSF IIS-1747614 and NSF IIS-2141037. Any opinions, findings, and conclusions or recommendations expressed in this material are those of the author(s) and do not necessarily reflect the views of the National Science Foundation.

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

## A  Related Work

**Model Compression-based Methods.** There are a number of well-explored model compression methods which can be used for accelerating model inference, such as weight quantization (Choukroun et al., 2019; Fan et al., 2020; Jin et al., 2021; Zhu et al., 2016), pruning (Zhu and Gupta, 2017), and knowledge distillation (Hinton et al., 2015). **Weight quantization** involves mapping model weights to low-precision integers and floating-point numbers, making them more hardware-friendly for computation. Specifically, (Xiao et al., 2022) proposed 8-bit quantization for BERT; (Tang et al., 2022) explored how to use 4 bits to quantize BERT; (Bai et al., 2020; Tian et al., 2023; Zhang et al., 2020) studied how to quantize BERT into 1-bit or 2-bits. **Pruning**, on the other hand, focuses on setting redundant parameters to zero to create a sparse network, enabling accelerated sparse matrix operations on specific hardware platforms. Existing pruning methods such as (Liu et al., 2021) and (Chen et al., 2020) proposed a dynamic structured pruning method and a lottery ticket hypothesis (Frankle and Carbin, 2018) based method respectively to learn a sparse network for BERT. **Knowledge distillation**, meanwhile, is to utilize a powerful large model (teacher model) to guide the learning of a lightweight model (student model). The lightweight student model usually has lower inference complexity compared to the teacher model. DistilBERT (Sanh et al., 2019), TinyBERT (Jiao et al., 2019), MobileBERT (Sun et al., 2020) and PKD (Sun et al., 2019) used knowledge distillation to learn a lightweight BERT. However, these compression-based methods have drawbacks. Firstly, they often require training compressed models from scratch, which can be computationally expensive. Secondly, weight quantization or pruning techniques rely on specialized hardware support, limiting their flexibility.

## B  Knowledge Distillation

Given a fine-tuned LM, i.e. $f_v(\cdot)$ learned in the first stage, knowledge distillation is to enable the proposed $f(\cdot)$ to mimic the prediction and hidden representation of $f_v$. Specifically, we consider the following losses: 1) The distillation loss of prediction is defined as

$$\mathcal{L}_p = \frac{1}{n} \sum_j \mathcal{D}_{KL}(f_v(x^i), f(x^i)), \qquad (5)$$

where $\mathcal{D}_{KL}(\cdot, \cdot)$ is the KL divergence between the probability over the two outputs. The loss $\mathcal{L}_p$ regu-

larizes the prediction of HadSkip to be consistent with that of $f_v$. 2) The distillation loss of hidden representations is to enables the hidden representations of $f$ and $f_v$ to be similar, which can be formulated as

$$\mathcal{L}_h = \frac{1}{nL} \sum_{j=1}^{n} \sum_{i=1}^{L} \text{MSE}(\boldsymbol{h}_i^j, {}_v\boldsymbol{h}_i^j), \qquad (6)$$

where $\boldsymbol{h}_i^j$, ${}_v\boldsymbol{h}_i^j$ are the $i$-th layer hidden representation of $f$ and $f_v$ with respect to input sequence $x_j$ respectively. 3) Similarly, the distillation loss of attention is to penalize the discrepancy between attention matrices of $f$ and $f_v$, which can be represented as

$$\mathcal{L}_{att} = \frac{1}{nL} \sum_{j=1}^{n} \sum_{i=1}^{L} \text{MSE}(\boldsymbol{A}_i^j, {}_v\boldsymbol{A}_i^j), \qquad (7)$$

where $\boldsymbol{A}_i^j$ and ${}_v\boldsymbol{A}_i^j$ are the $i$-th layer averaged attention matrices of $f$ and $f_v$ with respect to input sequence $x_j$ respectively. 4) The distillation loss of embedding layer is

$$\mathcal{L}_{emb} = \frac{1}{n} \sum_{j} \text{MSE}(\boldsymbol{e}^j, \boldsymbol{e}_v^j), \qquad (8)$$

where $\boldsymbol{e}^j$ and $\boldsymbol{e}_v^j$ are the embedding layer outputs of $f$ and $f_v$ with respect to input sequence $x_j$. In summary, the total loss function is obtained by combining it with the task-specific loss, which can be formulated as

$$\mathcal{L} = \frac{1}{n} \sum_{j} \ell(f(x^j; \Theta))$$
$$= \mathcal{L}_t + \alpha_1 \mathcal{L}_p + \alpha_2 \mathcal{L}_h + \alpha_3 \mathcal{L}_{att} + \alpha_4 \mathcal{L}_{emb},$$

where $\mathcal{L}_t = \frac{1}{n} \sum_{j} \ell_t(f(x^j; \Theta), y^j)$ is the task-specific loss function, and $\alpha_1, \alpha_2, \alpha_3, \alpha_4$ are hyperparameters.

## C  Algorithm of Reinmax

---
**Algorithm 1:** Reinmax

**Input:** gate function input $g(\boldsymbol{h}_{i-1}; \omega_i)$
**Output:** $\boldsymbol{B}[0]$

1   $\pi_0 = \text{Softmax}(g(\boldsymbol{h}_{i-1}; \omega_i))$;
2   $\boldsymbol{D} = \text{One\_Hot}(\pi_0)$;
3   $\pi_1 = \frac{\boldsymbol{D} + \pi_0}{2}$;
4   $\pi_1 = \text{Softmax}(\text{stop\_gradient}(\log(\pi_1) - g(\boldsymbol{h}_{i-1}; \omega_i)) + g(\boldsymbol{h}_{i-1}; \omega_i))$;
5   $\pi_2 = 2\pi_1 - \frac{1}{2}\pi_0$;
6   $\boldsymbol{B} = \pi_2 - \text{stop\_gradient}(\pi_2) + \boldsymbol{D}$;

---

| Dataset | Task | Metric |
|---------|------|--------|
| MRPC | Paraphrase Identification | F1 |
| QQP | Paraphrase Identification | F1 |
| MNLI | Natural Language Inference | Accuracy |
| QNLI | Natural Language Inference | Accuracy |
| RTE | Natural Language Inference | Accuracy |
| SST-2 | Sentiment Classification | Accuracy |
| CoLA | Linguistic Acceptability | Matthews corr |

Table 5: Dataset description.

## D  Dataset

## E  Implementation Details

We simply set $\alpha_1$, $\alpha_2$, $\alpha_3$, $\alpha_4$, and $\beta$ as 1 for all experiments. We select the learning rates from $\{5e-6, 1e-5, 2e-5, 3e-5, 5e-5\}$ and batch size from $\{8, 16, 32, 64\}$. Experiments are conducted on four NIVIDA RTX A6000.