# OpenReview forum: "HadSkip: Homotopic and Adaptive Layer Skipping of Pre-trained Language Models for Efficient Inference"
_EMNLP/2023/Conference — EMNLP 2023 Findings_

### Official Review · Reviewer_6pFJ · 2023-08-02

**Soundness:** 4

**Excitement:**

4: Strong: This paper deepens the understanding of some phenomenon or lowers the barriers to an existing research direction.

**Paper Topic And Main Contributions:**

This article studies how to perform skip-layer inference during pre-trained model inference, which is a significant improvement over the previous method of early exit. This work trains a gating network to determine whether each Transformer layer needs to be inferred and sets a maximum budget for inference layers. Experimental results show that under the same budget layer, the method proposed in this paper is significantly better than the baseline method.

**Questions For The Authors:**

- Issues in the **Reasons To Reject** Part
- Can the method be transferred to generative language models like GPT?



**Reasons To Accept:**

- The skip-level inference approach overcomes the problem of the previous early exit method, which can only perform fixed-order inference on the first k layers.

- A new gating network mechanism is proposed to control whether each layer needs to be inferred.

- Empirical results demonstrate the advantages of the proposed method over the baseline method.

- Detailed analytical experiments.

**Reasons To Reject:**

 - The new gating module introduces additional parameters and computational complexity, but the paper lacks analysis and comparison with the baseline in this regard.
- The paper only reports results on three datasets beyond Table 1, and it would be beneficial to see results on a more diverse range of datasets.
- The paper lacks an in-depth analysis of why skip-layer inference works, which could guide us in designing better methods for accelerating inference.





**Reproducibility:**

4: Could mostly reproduce the results, but there may be some variation because of sample variance or minor variations in their interpretation of the protocol or method.

**Reviewer Confidence:**

4: Quite sure. I tried to check the important points carefully. It's unlikely, though conceivable, that I missed something that should affect my ratings.

---

> ### Author Rebuttal · Authors · 2023-08-29
>
> >The new gating module introduces additional parameters and computational complexity, but the paper lacks analysis and comparison with the baseline in this regard.
>
> The extra parameters account for only about 6\% of the BERT-base, and the additional computational complexity (FLOPs) is quite small. The FLOPs of one transformer layer of BERT is about $24sH^{2}+4Hs^{2}$, the FLOPs of embedding layer is about $2sH$, and the FLOPs of output layer is about $2H^{2}k$, where $s$ is the sequence length, $H$ is the embedding dimension, and $k$ is the number of classes. For a model with $L$ layers, it uses $(24sH^{2}+4Hs^{2})L+2sH+2H^{2}k$ FLOPs for inference one time. For HadSkip, it needs $6H^{2}L$ FLOPs for gates. If HadSkip uses $l$ layers, it needs $(24sH^{2}+4Hs^{2})l+2sH+2H^{2}k+6H^{2}L$ FLOPs for inference one time, where $6H^{2}L$ is the additional computation for gates. In experiments, we use $s=128$. Thus, for a binary classification task, BERT-base needs about 22.3G FLOPs. The HadSkip uses about 5.6G FLOPs, 11.2G FLOPs, 16.8G FLOPs with respect to using 3 layers, 6 layers and 9 layers respectively. They roughly correspond to 1.3x, 2x, 4x inference acceleration. Therefore, compared to the transformer layers, the additional parameters and computational complexity are quite light.
>
> >The paper only reports results on three datasets beyond Table 1, and it would be beneficial to see results on a more diverse range of datasets.
>
> Thanks for your valuable suggestion. We conducted experiments using the GLUE benchmark, as recommended in prior research[1]. Due to resource constraints, we selected three representative tasks, namely MRPC, MNLI, and CoLA, for conducting experiments about ablation studies, extending to RoBERTa, and exploring combinations with TinyBERT. However, they have covered three typical tasks of natural language understanding. MNLI serves as a dataset for inference tasks, MRPC for paraphrase tasks, and CoLA for acceptability assessments. Both MNLI and MRPC involve sentence pairs, while CoLA focuses on the comprehension of individual sentences. Additionally, prior research has demonstrated a strong positive correlation between performance on MNLI and that on other understanding tasks[2]. Consequently, we present the model's performance on these three datasets. In future work, we can consider incorporating additional datasets.
>
> >The paper lacks an in-depth analysis of why skip-layer inference works, which could guide us in designing better methods for accelerating inference.
>
> As stated in [3], each layer of BERT acquires distinct information, with surface features at the lower levels, syntactic features in the middle, and semantic features at the top. Consequently, we contend that for varying input data, the layers playing significant roles exhibit diversity. This constitutes the motivation behind HadSkip's adaptive layer skipping mechanism. Furthermore, we have visualized the layer selection on various datasets in Figures 2-4 to validate our motivation. We posit that this discovery can offer valuable insights into the development of more effective approaches for accelerating inference, such as the adaptive selection of both model weights and layers.
>
> >Can the method be transferred to generative language models like GPT?
>
> This is a very good point! We are currently researching the extension of HadSkip to generative language models. Our preliminary findings suggest that HadSkip can skip a few layers with minimal performance degradation, for example, 2 or 3 layers on GPT2. Nonetheless, the exploration of how to skip more layers with minimal performance decline remains a subject of ongoing research. We intend to incorporate this aspect into our future work.
>
> [1] Zhang Z, Zhu W, Zhang J, et al. Pcee-bert: Accelerating bert inference via patient and confident early exiting[C]//Findings of the Association for Computational Linguistics: NAACL 2022. 2022: 327-338.
>
> [2]Pruksachatkun Y, Phang J, Liu H, et al. Intermediate-task transfer learning with pretrained models for natural language understanding: When and why does it work?[C]//58th Annual Meeting of the Association for Computational Linguistics, ACL 2020. Association for Computational Linguistics (ACL), 2020: 5231-5247.
>
> [3] Jawahar G, Sagot B, Seddah D. What does BERT learn about the structure of language?[C]//ACL 2019-57th Annual Meeting of the Association for Computational Linguistics. 2019.

---

### Official Review · Reviewer_ah3X · 2023-08-02

**Soundness:** 4

**Excitement:**

4: Strong: This paper deepens the understanding of some phenomenon or lowers the barriers to an existing research direction.

**Missing References:**

[1] Reducing Transformer Depth on Demand with Structured Dropout - Fan et al. ICLR 2020

**Paper Topic And Main Contributions:**

The paper introduces HadSkip for learning to adaptively skipping layers for fast pre-trained language model inference.
The paper does this through three stages.

Stage 1: The authors fine-tune the model in a vanilla manner.

Stage 2: The authors freeze the fine-tuned parameters. They add discrete binary gates for each layer. They train the gates using ReinMax.

Stage 3: The author jointly further trains both the gate and non-gate parameters through successive substages with different layer budget hyperparameter (motonologically decreasing - making the optimization successively harder). Knowledge distillation is also used for smoother optimization.

Outperforms many comparable methods - particularly different early exit strategies with BERT on GLUE tasks.

**Questions For The Authors:**

I am open to rebuttal/discussion related to point 2. That's my main concern. I am willing to increase my score if this is addressed satisfactorily.

**Reasons To Accept:**

1. To my knowledge seems like a fairly novel method and overall an interesting and reasonable combination of techniques.

2. Decent performance compared to early exit methods and nice analysis.

**Reasons To Reject:**

1. Empirical time-memory savings seem missing. FLOPs could be good.
2. A very relevant simple baseline seems missing - LayerDrop [1]. LayerDrop applies simple dropout at the level of layers (related to BlockDrop cited in the paper) and achieves ability to prune a high number of layers during inference. The paper shows promising performance on RoBERTa - often even beating the non-pruned vanilla baseline. In contrast, the introduced method here is much more complicated to implement and performs slightly worse than vanilla. This seems to be a critical miss.

[1] Reducing Transformer Depth on Demand with Structured Dropout - Fan et al. ICLR 2020

Post-Rebuttal:

My concerns are addressed. I am raising the soundness score to 4.

**Reproducibility:**

3: Could reproduce the results with some difficulty. The settings of parameters are underspecified or subjectively determined; the training/evaluation data are not widely available.

**Reviewer Confidence:**

3: Pretty sure, but there's a chance I missed something. Although I have a good feel for this area in general, I did not carefully check the paper's details, e.g., the math, experimental design, or novelty.

---

> ### Author Rebuttal · Authors · 2023-08-29
>
> >Empirical time-memory savings seem missing. FLOPs could be good.
>
> The proposed HadSkip can accelerate model inference. The FLOPs of one transformer layer of BERT is about $24sH^{2}+4Hs^{2}$, the FLOPs of embedding layer is about $2sH$, and the FLOPs of output layer is about $2H^{2}k$, where $s$ is the sequence length, $H$ is the embedding dimension, and $k$ is the number of classes. For a model with $L$ layers, it requires $(24sH^{2}+4Hs^{2})L+2sH+2H^{2}k$ FLOPs for a single inference. For HadSkip, it needs $6H^{2}L$ FLOPs for gates. If HadSkip uses $l$ layers, it needs $(24sH^{2}+4Hs^{2})l+2sH+2H^{2}k+6H^{2}L$ FLOPs for inference one time. In experiments, we use $s=128$. Thus, for a binary classification task, BERT-base needs about 22.3G FLOPs. The HadSkip uses about 5.6G FLOPs, 11.2G FLOPs, 16.8G FLOPs with respect to using 3 layers, 6 layers and 9 layers respectively. They roughly correspond to 1.3x, 2x, 4x inference acceleration.
>
> >A very relevant simple baseline seems missing - LayerDrop [1]. LayerDrop applies simple dropout at the level of layers (related to BlockDrop cited in the paper) and achieves ability to prune a high number of layers during inference. The paper shows promising performance on RoBERTa - often even beating the non-pruned vanilla baseline. In contrast, the introduced method here is much more complicated to implement and performs slightly worse than vanilla. This seems to be a critical miss.
>
> Thanks for pointing out it! In fact, our proposed HadSkip differs significantly from LayerDrop. **First, their settings are different.** LayerDrop incorporates layer skipping during the pre-training stage and subsequently fine-tunes the model for downstream tasks. In contrast, our proposed HadSkip does not include the pre-training stage. The pre-training stage is resource-intensive, and it is impractical to re-train every pre-trained language model upon release. Therefore, HadSkip is designed under a more realistic setting. **Secondly, LayerDrop is not an input-adaptive method, whereas our proposed HadSkip can dynamically skip layers based on different inputs.** Additionally, with LayerDrop, there are instances where it outperforms a smaller RoBERTa model with the same number of layers as LayerDrop, rather than outperforming the vanilla RoBERTa. To facilitate a more direct comparison with LayerDrop, we have excluded the knowledge distillation module and present the performance of HadSkip using RoBERTa as the backbone, as displayed in Table A. It is evident that, even without additional pre-training, our proposed HadSkip can achieve comparable performance to LayerDrop and even outperform it when utilizing 6 layers on MNLI.
>
> Table A: Performance comparison using RoBERTa as the backbone. To make a fair comparison, we remove KD used in HadSkip. Results of LayerDrop are taken from LayerDrop paper directly.
> | | Budget | MNLI | SST2 |
> |-----------|------|------|------|
> |LayerDrop| 3      | 78.6 | 90.5 |
> |                 | 6      |82.9 | 92.5 |
> |HadSkip w/o KD| 3      | 78.4 | 90.7 |
> |                           | 6     | 85.1 | 92.5 |

---

### Official Review · Reviewer_a8ng · 2023-08-05

**Soundness:** 2

**Excitement:**

2: Mediocre: This paper makes marginal contributions (vs non-contemporaneous work), so I would rather not see it in the conference.

**Paper Topic And Main Contributions:**

This paper proposes an adaptive layer-skipping method for Pre-trained Transformer models (BERT, RoBERTa, etc.). HadSkip introduces the gate network at each layer to skip the layer or not. By training with knowledge distillation and homotopic optimization, they improve previous early exiting methods on various backbones and datasets.

**Reasons To Accept:**

1. The layer-skipping method is easy to follow and can combine with other model acceleration methods.
2. HadSkip outperforms previous early exiting methods on various backbones and datasets.

**Reasons To Reject:**

1. The novelty is limited. Although HadSkip focuses on improving early exiting, it also can be seen as a dynamic neural network an various methods have been proposed for Transformers [1]. The overall method is overly complex compared to baselines, and the introduction of tricks such as KD and Homotopic Optimization Strategy (**not used in baselines**, such as PABEE) does not clearly reflect the gain of the method itself.
2. The authors only report efficiency metrics by budget (3/6/9 layers), while for an efficient method of NLP, they should also report the speed-up of FLOPs or latency (which is more important for real life.). By the way, because HadSkip uses knowledge distillation when compared with some early exiting baselines, the KD contribution should also be considered.
3. Lack of some baseline comparisons. Layerdrop [2] and dynamic token pruning should also be compared. LayerDrop share a similar idea with HadSkip. Dynamic token pruning also can be seen as a fine-grained early exiting method [3]. In addition, the authors need to clearly identify whether they will **open source** code or not.

[1] Dynamic Neural Networks: A Survey. Han et al. 2021.

[2] Reducing Transformer Depth on Demand with Structured Dropout. Fan et al. 2019.

[3] Transkimmer: Transformer Learns to Layer-wise Skim. Guan et al. 2022.

**Reproducibility:**

2: Would be hard pressed to reproduce the results. The contribution depends on data that are simply not available outside the author's institution or consortium; not enough details are provided.

**Reviewer Confidence:**

4: Quite sure. I tried to check the important points carefully. It's unlikely, though conceivable, that I missed something that should affect my ratings.

---

> ### Author Rebuttal · Authors · 2023-08-29
>
> >The novelty is limited. Although HadSkip focuses on improving early exiting, it also can be seen as a dynamic neural network an various methods have been proposed for Transformers [1]. The overall method is overly complex compared to baselines, and the introduction of tricks such as KD and Homotopic Optimization Strategy (not used in baselines, such as PABEE) does not clearly reflect the gain of the method itself.
>
> Firstly, it's important to note that the term "dynamic neural network" encompasses a wide range of methods. In the context of accelerating language model inference, our proposed HadSkip stands out as notably distinct from three common dynamic neural networks: early exiting, mixture of experts (MoE), and token pruning. In the case of early exiting-based methods, they necessitate the input to traverse the transformer layers sequentially before exiting. Conversely, our proposed HadSkip exhibits the capability to dynamically select layers based on the current input, thus offering greater flexibility. In contrast to MoE, the proposed HadSkip is notably more straightforward to train due to its ability to mitigate challenges related to load imbalance and under-utilization of experts[a]. Token pruning methods can be categorized as unstructured pruning methods, making it challenging to achieve substantial acceleration without specialized hardware. In contrast, the proposed HadSkip skips multiple layers in a structured manner, rendering it more feasible for implementation on general hardware. **Therefore, while both these methods and our proposed HadSkip fall under the category of dynamic neural networks, it is evident that HadSkip is distinct from existing approaches. Moreover, beyond the disparities in model architectures, we introduce a novel three-stage and homotopic optimization strategy, which represents a substantial departure from existing methodologies.**
>
> Secondly, we have conducted an ablation study to investigate the impact of KD and the homotopic optimization strategy, as detailed in Table 2 of Section 5.3. Based on the findings presented in Table 2, it is evident that our proposed HadSkip consistently outperforms the baselines even in the absence of KD or the homotopic optimization strategy. Additionally, the proposed homotopic optimization strategy aims to enhance the stability of gating, constituting a crucial component of our approach. Consequently, through a comparative analysis with the baselines in Section 5.2 and the ablation study in Section 5.3, we can establish the effectiveness of our proposed HadSkip.
>
> >The authors only report efficiency metrics by budget (3/6/9 layers), while for an efficient method of NLP, they should also report the speed-up of FLOPs or latency (which is more important for real life.). By the way, because HadSkip uses knowledge distillation when compared with some early exiting baselines, the KD contribution should also be considered.
>
> The FLOPs of one transformer layer of BERT is about $24sH^{2}+4Hs^{2}$, the FLOPs of embedding layer is about $2sH$, and the FLOPs of output layer is about $2H^{2}k$, where $s$ is the sequence length, $H$ is the embedding dimension, and $k$ is the number of classes. For a model with $L$ layers, it requires $(24sH^{2}+4Hs^{2})L+2sH+2H^{2}k$ FLOPs for one single inference. For HadSkip, it needs $6H^{2}L$ FLOPs for gates. If HadSkip uses $l$ layers, it needs $(24sH^{2}+4Hs^{2})l+2sH+2H^{2}k+6H^{2}L$ FLOPs for a single inference. In experiments, we use $s=128$. Thus, for a binary classification task, BERT-base needs about 22.3G FLOPs. The HadSkip uses about 5.6G FLOPs, 11.2G FLOPs, 16.8G FLOPs with respect to using 3 layers, 6 layers and 9 layers respectively. They roughly correspond to 1.3x, 2x, 4x inference acceleration.
>
> >Lack of some baseline comparisons. Layerdrop [2] and dynamic token pruning should also be compared. LayerDrop share a similar idea with HadSkip. Dynamic token pruning also can be seen as a fine-grained early exiting method [3]. In addition, the authors need to clearly identify whether they will open source code or not.
>
> First, the setting of Layerdrop is different from ours. Layerdrop trains layer skipping during the pre-training stage and subsequently fine-tunes the model for downstream tasks. In contrast, our proposed HadSkip does not incorporate the pre-training stage. The pre-training stage is costly, and re-training each pre-trained language model upon release is impractical. Consequently, HadSkip is designed within a more practical framework. Moreover, Layerdrop cannot adaptively drop layers for individual input data, whereas HadSkip can dynamically select layers. We include the performance results of the proposed HadSkip(w/o KD) based on RoBERTa in Table A to demonstrate its superiority over Layerdrop as well. According to Table A, we observe that despite not conducting additional pre-training, our proposed HadSkip can achieve comparable performance to Layerdrop and even outperform it when using 6 layers on MNLI.
>
> As for Transkimmer, we believe that comparing it is beyond the scope of our study. Transkimmer is a token pruning method that modifies the model input but does not alter the model architecture. In contrast, HadSkip employs distinct model architectures for different inputs while keeping the input unchanged. The two methods are orthogonal[b]. Consequently, we have selected early exiting-based methods that are more closely related as our baselines. In future work, we intend to investigate the performance of combining HadSkip with Transkimmer. We plan to release our code in the camera-ready version to further promote reproducibility.
>
> Table A: Performance comparison using RoBERTa as the backbone. To make a fair comparison, we remove KD used in HadSkip. Results of LayerDrop are taken from LayerDrop paper directly.
> | | Budget | MNLI | SST2 |
> |-----------|------|------|------|
> |LayerDrop| 3      | 78.6 | 90.5 |
> |                 | 6      |82.9 | 92.5 |
> |HadSkip w/o KD| 3      | 78.4 | 90.7 |
> |                           | 6     | 85.1 | 92.5 |
>
> [a] Zuo S, Zhang Q, Liang C, et al. MoEBERT: from BERT to Mixture-of-Experts via Importance-Guided Adaptation[C]//Proceedings of the 2022 Conference of the North American Chapter of the Association for Computational Linguistics: Human Language Technologies. 2022.
>
> [b] Xu C, Mcauley J. A Survey on Dynamic Neural Networks for Natural Language Processing[C]//Findings of the Association for Computational Linguistics: EACL 2023. 2023: 2325-2336.

---

### Meta-Review · Area_Chair_fMXR · 2023-09-19

**Recommendation:** 5

**Metareview:**

The paper presents a skip-computing model which achieved good results on basic benchmark dataset. It's a well written paper and demonstrate good empirical performance.

Pros:

1. Review work is solid
2. An effective method.
3. Write well and easy to follow.

Cons:

1. The comparison baseline is a bit limited. Despite during discussion period authors add many results and commit they will add to the final version, there is no guarantee this will be happening and the additional information could be integrated well into present materials.

2. Authors are not proactive on adding real-time wall-clock results. Only after many rounds of discussion, they add some statistics compared to Transkimmer only. And it's only speed-up ratio but not wall-clock time which makes the results not verifiable.

3. The comparing models are relatively small.

---

### Decision · Program_Chairs · 2023-10-07

**Decision:**

Accept-Findings

**Comment:**

The paper presents a skip-computing model which achieved good results on basic benchmark dataset. It's a well written paper and demonstrate good empirical performance.

Pros:

1. Review work is solid
2. An effective method.
3. Write well and easy to follow.

Cons:

1. The comparison baseline is a bit limited. Despite during discussion period authors add many results and commit they will add to the final version, there is no guarantee this will be happening and the additional information could be integrated well into present materials.

2. Authors are not proactive on adding real-time wall-clock results. Only after many rounds of discussion, they add some statistics compared to Transkimmer only. And it's only speed-up ratio but not wall-clock time which makes the results not verifiable.

3. The comparing models are relatively small.